# Recruitment of inhibition and excitation across mouse visual cortex depends on the hierarchy of interconnecting areas

**Rinaldo David D'Souza[1]\*, Andrew Max Meier[1], Pawan Bista[1], Quanxin Wang[2], Andreas Burkhalter[1]\***

[1]Department of Neuroscience, Washington University School of Medicine, St. Louis, United States; [2]Allen Institute for Brain Science, Seattle, United States

**Abstract** Diverse features of sensory stimuli are selectively processed in distinct brain areas. The relative recruitment of inhibitory and excitatory neurons within an area controls the gain of neurons for appropriate stimulus coding. We examined how such a balance of inhibition and excitation is differentially recruited across multiple levels of a cortical hierarchy by mapping the locations and strengths of synaptic inputs to pyramidal and parvalbumin (PV)-expressing neurons in feedforward and feedback pathways interconnecting primary (V1) and two higher visual areas. While interareal excitation was stronger in PV than in pyramidal neurons in all layer 2/3 pathways, we observed a gradual scaling down of the inhibition/excitation ratio from the most feedforward to the most feedback pathway. Our results indicate that interareal gain control depends on the hierarchical position of the source and the target, the direction of information flow through the network, and the laminar location of target neurons.

\*For correspondence: dsouzar@ wustl.edu (RDD); burkhala@wustl. edu (AB)

**Competing interests:** The authors declare that no competing interests exist.

## Introduction

Visual perception and visually guided actions result from the coordinated neuronal communication between multiple, functionally diverse areas of visual cortex. Within visual cortex, interareal communication is achieved through the axons of pyramidal (Pyr) cells carrying feedforward (FF) information from lower to higher areas and feedback (FB) signals through 'top-down' connections descending across the hierarchy of visual areas (*Felleman and Van Essen, 1991*; *Coogan and Burkhalter, 1993*). How neurons within such a highly interconnected network and increasing densities of inputs at higher levels of the cortical hierarchy (*Wang et al., 2012*; *Elston, 2003*) maintain stimulus-specificity without saturating their spike output has been studied by modelling the effects of inhibitory synaptic inputs and by recording the balance of excitation and inhibition in local networks of sensory cortex (*Shadlen and Newsome, 1998*; *Pouille et al., 2009*). However, the rules by which the inhibition/excitation (I/E) balance changes along processing pathways from early to deep stages of the brain and back are incompletely understood.

In the rodent visual system, interareal FF and FB pathways communicate through excitatory synapses contacting Pyr and GABAergic neurons (*Johnson and Burkhalter, 1996*). In the target area, both cell types are reciprocally connected by a fine-scale circuit embedded within the global network (*Yoshimura and Callaway, 2005*; *Jiang et al., 2015*; *Pfeffer et al., 2013*). Although interareal FF and FB connections terminate on multiple types of GABAergic neurons, most of them synapse onto PV-expressing fast-spiking interneurons (*Gonchar and Burkhalter, 1999*; *Gonchar and Burkhalter, 2003*; *Hangya et al., 2014*), which provide feedforward inhibition (FFI) to local Pyr cells (*Dong et al., 2004*; *Yang et al., 2013*).

**eLife digest** The visual cortex is the part of the brain responsible for the conscious sense of vision. It is made up of multiple connected areas, and each area has a different expertise for analyzing images. The areas exchange information about the outside world via connections between cells called neurons. Communication between the areas works like a hierarchy with deeper, more connected areas in the brain extracting more complex information from a visual scene.

Communication in the cortex requires repeated stimulation or "excitation" of pathways of neurons; this risks damage or loss of sensitivity. But all of the communication in the hierarchy is excitatory, meaning that a signal from one area activates other areas in the visual cortex. So, how does the brain avoid becoming over-stimulated? The answer is that connections between the areas of the visual cortex also contact inhibitory neurons that suppress brain activity. However, it is not clear how the level of inhibition in different areas of the visual cortex is fine-tuned to avoid over-stimulation while maintaining accurate perception of vision.

D'Souza et al. now report how three distinct areas of the mouse visual cortex communicate to process visual signals. The approach involved making particular pathways of neurons sensitive to light, such that they could be activated separately with a laser. Next, D'Souza et al. measured the activity of both inhibitory and excitatory neurons that link the different brain areas. The experiments showed that the inhibitory neurons are more strongly activated in the areas of the brain that are further up the hierarchy. This indicates that our ability to make sense of more complex features of visual signals requires higher levels of inhibitory control. The next step is to examine how the brain activates and controls inhibitory neurons, and how this depends on the situation an animal is in and the task it is performing.

FFI is a common functional motif throughout the brain, capable of regulating the I/E balance and thereby influencing the gain, the integration window, and the temporal precision of inputs (*Shadlen and Newsome, 1998*; *Atallah et al., 2012*; *Gabernet et al., 2005*; *Cardin et al., 2010*). Similar to the thalamocortical and local circuits in mouse barrel cortex and in V1 (*Atallah et al., 2012*; *Gabernet et al., 2005*), FFI is also involved in interareal communication across the visual cortical hierarchy (*Dong et al., 2004*). In fact, our studies in mouse visual cortex have shown that FF input to Pyr cells is more strongly counterbalanced by inhibition than FB input, suggesting pathway-specific differences in the gain and dynamic range of recurrent excitation involved in cortical computations (*Yang et al., 2013*; *Atallah et al., 2012*; *Okun and Lampl, 2008*).

Here, we demonstrate that higher relative inhibition in FF than in FB pathways is part of a more general rule of cortico-cortical communication. We studied the strengths of FF and FB inputs interconnecting mouse V1 with Pyr and PV neurons in the extrastriate area PM (posteromedial) situated high in the hierarchy, and compared the I/E balance with input to and from the hierarchically intermediate area LM (lateromedial). Using whole-cell patch clamp recordings and laser-scanning photostimulation of Channelrhodopsin-2 (ChR2)-expressing FF and FB connections in acute cortical slices, we show that the relative strength of excitatory input to Pyr and PV cells is pathway-specific and depends on the position of the source and target areas within the hierarchy. Interareal inputs to PV interneurons in upper layers, but not in lower layers, are stronger than to Pyr cells, and the asymmetry of I/E balance was greater for V1 to PM than for LM to PM connections, suggesting weaker FFI by inputs from hierarchically higher areas. In support of the notion that pathways from sources deeper in the brain, which deliver input that vary over a narrower range than input from the outside world, would require lower levels of inhibitory control, we found that FFI is weaker in FB connections and weakest in FB input to V1, at the bottom of the hierarchy. Our findings therefore suggest that in FF and FB pathways targeting neurons in layer 2/3 (L2/3), excitation is more strongly counterbalanced by inhibition and that the imbalance is gradually rectified according to hierarchical distance from the most feedforward to the most feedback.

## Results

### Hierarchy between V1, LM and PM

To study interareal inhibition across different levels of the cortical architecture, we first asked whether the visual areas V1, LM and PM lie at distinct levels of an interconnected hierarchical network. To do this we traced the outputs from V1, LM and PM with the anterograde tracer biotinylated dextran amine (BDA) and studied the laminar patterns of axon terminals in each of the visual cortical target areas: V1, LM, POR (postrhinal), AL (anterolateral), P (posterior), LI (laterointermediate), PM, AM (anteromedial), RL (rostrolateral) and A (anterior) (*Figure 1*, *Figure 1—figure supplement 1–3*). Projections were assigned to areas by their locations relative to retrogradely bisbenzimide-labeled callosal landmarks (*Wang and Burkhalter, 2007*), which we imaged in situ before sectioning the brain, and by their relative positions to each other (*Figure 1a*). Sections were numbered from the posterior pole of cortex so that the callosal landmarks seen in the coronal plane could be matched to specific locations of the in situ pattern. BDA labeled fibers were then superimposed onto the callosal pattern observed in the same section, and projections were assigned to specific areas according to the map by Wang and Burkhalter (*Wang and Burkhalter, 2007*) (*Figure 1a*). Optical density maps of projections showed striking laminar differences (*Figure 1b*). Although most projections involved L1-6, inputs from V1 consistently showed dense terminations in L2-4 of each of the higher areas with much sparser projections in L1. In contrast, projections from both LM and PM strongly targeted L1 of V1 while weakly targeting L2-4 (*Figure 1—figure supplement 2* and *3*). The selective targeting of L1 by FB projections is consistent with observations in other species (*Felleman and Van Essen, 1991*; *Coogan and Burkhalter, 1993*; *Rockland and Virga, 1989*; *Henry et al., 1991*). To analyze these patterns quantitatively, we computed the density ratio (DR) of terminations in L2-4 to that in L1 of axons from V1, LM and PM to each of the other nine areas and plotted DRs in a $3 \times 9$ matrix (*Figure 1c*). We reasoned that FF projections from lower areas would, on average, have a higher DR than FB projections from higher areas. The matrix showed that the average DRs in all targets of V1 were >2.52 ± 0.31, whereas the DRs for projections to V1 were <0.72 ± 0.08 (*Figure 1c*). Pairwise comparisons of average DRs of projections from each of V1, LM, and PM to the other areas showed significant (p<0.001, Mann-Whitney U test) differences, demonstrating that V1, LM and PM are at distinct hierarchical levels, with LM at the intermediate level between V1 and PM (*Figure 1d,e*).

### L2/3 FF and FB pathways between V1 and PM

Cortico-cortical inhibition between areas involves both, the initial excitation of interneurons by long-range axonal projections of Pyr cells, and the disynaptic inhibition of Pyr cells by interneurons. As a first step in the analysis of the recruitment of interareal inhibition, we first confirmed the role of PV interneurons in inhibiting neighboring Pyr cells. These experiments were performed in area PM, one of the targets of V1, in acute slices from mice in which PV cells expressed tdTomato (tdT) (*Figure 2—figure supplement 1a*). The axonal projections from V1 to PM were labeled by anterograde tracing with adeno associated virus (AAV) expressing a ChR2-Venus fusion protein (*Petreanu et al., 2009*) (*Figure 2—figure supplement 1a–f*). PM is the posterior projection zone medial to the densely type 2 muscarinic acetylcholine receptor (M2)-expressing area V1 (*Wang et al., 2011*; *Ji et al., 2015*) (*Figure 2—figure supplement 1d,e*). Similar to other cortical areas, PM contained tdT-PV cell bodies in L2-6, with axons and dendrites reaching into L1 (*Figure 2—figure supplement 1f*, *Figure 2—figure supplement 2a*). We performed paired recordings to examine whether increasing the excitation of PV cells results in stronger inhibition of neighboring synaptically connected Pyr cells (*Figure 2—figure supplement 2c*). To do this, we evoked action potentials by injecting current steps (100, 200, 300, and 400 pA; 50 ms; *Figure 2—figure supplement 2c–f*) into PV cells and recorded inhibitory postsynaptic currents (IPSCs) in connected Pyr cells. Similar to recordings in other cortical areas (*Pfeffer et al., 2013*; *Packer and Yuste, 2011*) we found a high connection probability. Recordings from PV and Pyr neurons within ~100 µm of each other resulted in 11/13 (84.6%) and 7/15 (46.6%) synaptically connected pairs in L2/3 and L5, respectively (*Figure 2—figure supplement 2g*). In both layers, increasing the firing of PV cells resulted in larger IPSCs in Pyr cells (n = 11 pairs in L2/3, 7 pairs in L5; *Figure 2—figure supplement 2d–f*). The increase in inhibition was due to both, the increased probability of PV cells to reach spike threshold, as well as increased spiking.

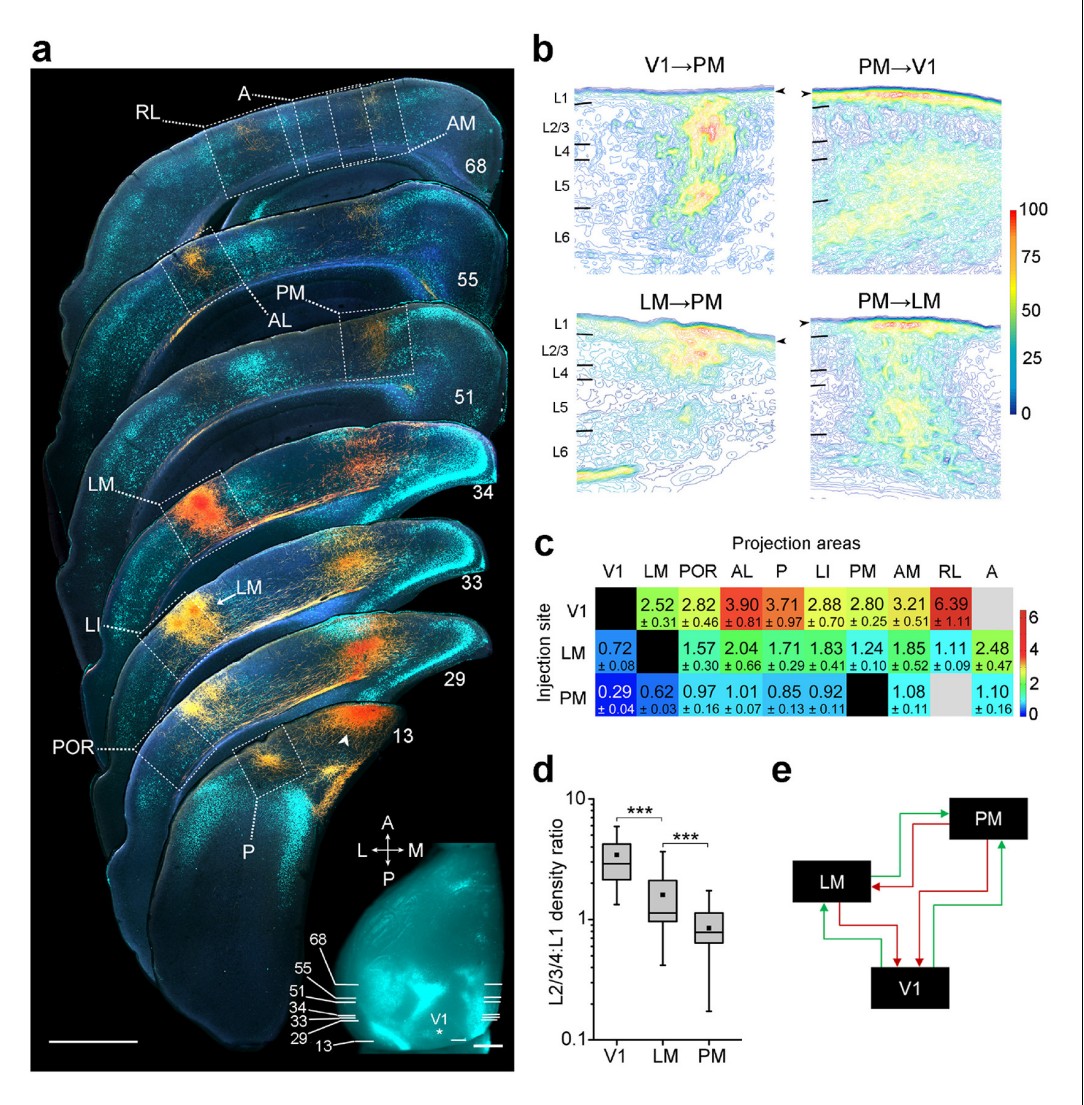

**Figure 1.** Hierarchy between V1, LM, and PM. (a) Rostrocaudal series of coronal slices through the left hemisphere showing anterogradely labelled axonal projections (yellow/orange) after V1 was injected with BDA. Retrogradely labelled callosally projecting neurons (light cyan), upon injection of bisbenzimide in the right hemisphere, act as landmarks for identification of areas (*Wang and Burkhalter, 2007*). Numbers denote sections corresponding to the positions shown in inset. See *Figure 1—figure supplement 1* for higher magnification of areas within dotted squares. Projection to LM adjacent to LI in section 33 is indicated. Arrowhead indicates a region in V1 near the injected site. In situ image of retrograde bisbenzimide-labelled callosally projecting neurons in the left hemisphere. Injection site in V1 (asterisk) and positions of coronal slices shown above are indicated. Scale bars, 1 mm. (b) Optical density of axonal projections in the target areas of the indicated pathways, normalized to peak density. Contours connect regions with similar optical densities. Arrowheads denote the edge of the slice and edge artifacts due to interpolation of optical density with dark background. (c) Color-coded heat map of L2-4:L1 density ratio (DR) for each of 25 distinct cortico-cortical connections. Blocks in grey indicate projections that were too weak to analyze. V1 exhibits the highest DRs, and PM the lowest, indicating the relative hierarchical positions of the areas. (d) The mean DR for all target areas is highest for V1, intermediate for LM, and lowest for PM; ***p<0.001, Mann-Whitney U-test. (e) Our schematic interpretation of the hierarchy of V1, LM, and PM in visual processing. Feedforward pathways are denoted in green, feedback in red.

The following figure supplements are available for figure 1:

**Figure supplement 1.** Darkfield images of the termination patterns of BDA-labelled axonal projections from V1 to LM, LI, P, POR, AL, PM, RL, AM and A.

*Figure 1 continued on next page*

*Figure 1 continued*

**Figure supplement 2.** Coronal sections showing anterogradely labelled axons (yellow/orange) from LM to V1, LI, P, POR, AL, PM, RL, AM and A, upon BDA injection into LM.

**Figure supplement 3.** Axonal projections (yellow/orange) from PM to V1, LM, LI, P, POR, AL, RL, AM and A, upon BDA injection into PM.

These findings reveal a local subnetwork that is likely tapped by interareal connections for FFI of Pyr cells in target areas.

Because the level of PV cell excitation determines the feedforward inhibitory drive to synaptically connected Pyr cells, we examined the strength of excitatory inputs to neighboring PV and Pyr cells by different pathways. We performed subcellular ChR2-assisted circuit mapping (sCRACM) (*Yang et al., 2013*; *Petreanu et al., 2009*; *Mao et al., 2011*) in acute slices of visual cortex to measure the input strength and the laminar location of interareal connections to PV and Pyr cells in different pathways. To study connections in the $FF_{V1 \rightarrow PM}$ pathway, we expressed ChR2-Venus in axons projecting from V1 to PM, and recorded excitatory postsynaptic currents (EPSCs) from PV and Pyr cells centered at the peak of the PM projection (*Figure 2a*). Photostimulation of ChR2-expressing axon terminals was achieved by a 473 nm laser delivered one spot at a time in a grid pattern separated by 75 µm (*Figure 2b*). Recordings were performed in the presence of 1 µm TTX (tetrodotoxin) and 50 µm 4-AP (4-aminopyridine) in the bath to block polysynaptic currents and repolarization of axon terminals, respectively. Resulting EPSCs were measured by whole cell patch clamp recordings from PV and Pyr neurons voltage-clamped at −70 mV (*Figure 2c*). We compared EPSCs between PV and Pyr neurons whose cell bodies were in the same layer of the same slice, within ~100 µm of each other.

In the L2/3 $FF_{V1 \rightarrow PM}$ pathway, EPSCs recorded from PV cells were larger than those from Pyr cells (*Figure 2c–f*). On average, the largest EPSCs were evoked from synaptic inputs to proximal dendrites at the bottom of L2/3 whereas inputs to distal dendrites were weaker (*Figure 2d,e*). The mean total current in PV cells was 12.85 ± 4.48-fold stronger than that in neighboring Pyr cells (p<0.001, n = 14 pairs). To illustrate the relative excitation of PV and Pyr cells, we plotted the total EPSC in each PV cell against the total EPSC in its Pyr neighbor, and measured the mean slope for all such pairs in this pathway (*Figure 2f*). We also computed the mean slope after normalizing the EPSC to the mean cell conductance to control for cell size. Similar to observations in thalamocortical and local circuits, the time to peak of EPSCs was significantly shorter in PV than in Pyr cells (*Figure 2g*; n = 14 pairs, p<0.05, paired t-test), consistent with the notion that PV interneurons can be recruited more rapidly than Pyr neurons in diverse brain areas (*Hull et al., 2009*; *Povysheva et al., 2006*).

We next asked whether connections to PV and Pyr cells in the $FB_{PM \rightarrow V1}$ pathway showed a different I/E balance. Recordings in V1 showed that similar to PM, EPSCs in PV cells were larger and faster than in Pyr cells (*Figure 3a–f*). In contrast to the $FF_{V1 \rightarrow PM}$ pathway, however, the excitation of PV cells in the $FB_{PM \rightarrow V1}$ pathway was only 1.93 ± 0.44-fold stronger than Pyr cell excitation. Thus, the excitation of PV cells, relative to that of neighboring Pyr cells, was weaker in the $FB_{PM \rightarrow V1}$ than in the $FF_{V1 \rightarrow PM}$ pathway (*Figure 3g*; n = 14 pairs for $FF_{V1 \rightarrow PM}$, n = 21 pairs for $FB_{PM \rightarrow V1}$; p<0.001), similar to previous observations in the L2/3 $FF_{V1 \rightarrow LM}$ and $FB_{LM \rightarrow V1}$ pathways (*Yang et al., 2013*). The larger EPSCs in PV cells could be a result of either a higher density of excitatory input or due to a larger area over which individual PV cells are contacted by interareal projections, or both. We therefore measured the mean EPSC per pixel and the total area over which each cell type exhibited measurable EPSCs (*Figure 3—figure supplement 1*). In the $FF_{V1 \rightarrow PM}$ pathway, PV cells exhibited larger EPSCs per pixel than Pyr cells (*Figure 3—figure supplement 1a*) as well as received input over a larger area (*Figure 3—figure supplement 1b*). In contrast, in the $FB_{PM \rightarrow V1}$ pathway, the mean EPSC per pixel between the two cell types were not significantly different (*Figure 3—figure supplement 1c*), indicating that the larger total EPSCs in PV cells were the result of PV cells receiving excitatory input over a larger area (*Figure 3—figure supplement 1d*).

The laminar organization of interareal input to individual neurons was significantly different for the two pathways. Unlike $FF_{V1 \rightarrow PM}$ projections, FB axons from PM provided strong inputs to L1 of V1. We quantified L1 input by measuring the total pixel values in each row of the photostimulation

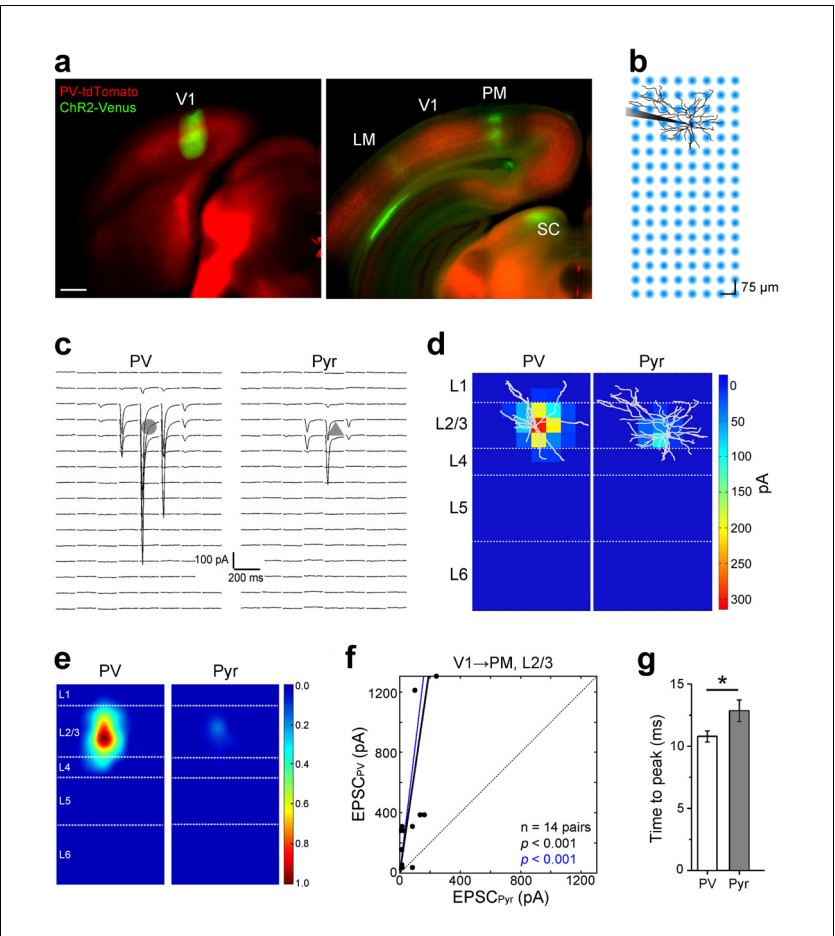

**Figure 2.** Subcellular ChR2-assisted mapping of V1→PM connections to L2/3 PV and Pyr cells. (**a**) Coronal slices showing injection (left) and target (right) sites two weeks after the injection of AAV2/1.CAG.ChR2-Venus.WPRE. SV40 into V1. Scale bar, 500 µm. Select target areas indicated in right panel. SC, superior colliculus. (**b**) Schematic of laser-scanning photostimulation of ChR2-expressing axon terminals during whole-cell recording of a biocytin-filled neuron. TTX and 4-AP are added to the bath solution, and the blue laser is delivered successively one spot at a time in a grid pattern separated by 75 µm. (**c**) EPSCs$_{sCRACM}$ in a PV (left) and a neighboring Pyr (right) cell upon photostimulation. Grey shapes denote the location of the cell body of the recorded neuron. (**d**) Heat map of mean EPSCs within 75 µs after photostimulation for the EPSCs in *3c*. Reconstructions of respective biocytin-filled neurons are superimposed on heat map. (**e**) Average heat map of 14 neighboring PV-Pyr cell pairs in L2/3 receiving V1→PM input, normalized to largest pixel value between a pair. PV cells receive substantially stronger input. (**f**) Scatter plot denoting the relative input strengths to 14 PV-Pyr cell pairs. Each data point represents a pair with the respective EPSCs in the PV (vertical axis) and the Pyr (horizontal axis) cell. The total EPSC in PV cells is significantly larger than that in neighboring Pyr cells ($p < 0.001$, Wilcoxon signed-rank test). Solid black line: mean slope, blue line: mean slope after normalizing currents to mean cell conductance. (**g**) The mean time to peak of EPSCs after photostimulation is larger in Pyr cells than in PV cells (*$p < 0.05$, paired t-test).

The following figure supplements are available for figure 2:

**Figure supplement 1.** V1→PM pathway in a PV-tdT mouse.

**Figure supplement 2.** Paired recordings of excitation-dependent, PV cell-mediated inhibition of Pyr cells.

grid pattern and plotting EPSCs against distance from the pial surface (*Figure 3h,i*). The values are percentages in each row of the total EPSC in the respective cell type. Consistent with the distribution of projections (*Figure 1b*) the proportion of inputs to L1, relative to total EPSCs, was larger in the FB$_{PM→V1}$ pathway than in the FF$_{V1→PM}$ pathway (*Figure 3i*; $p < 0.01$ for both PV and Pyr cells). It

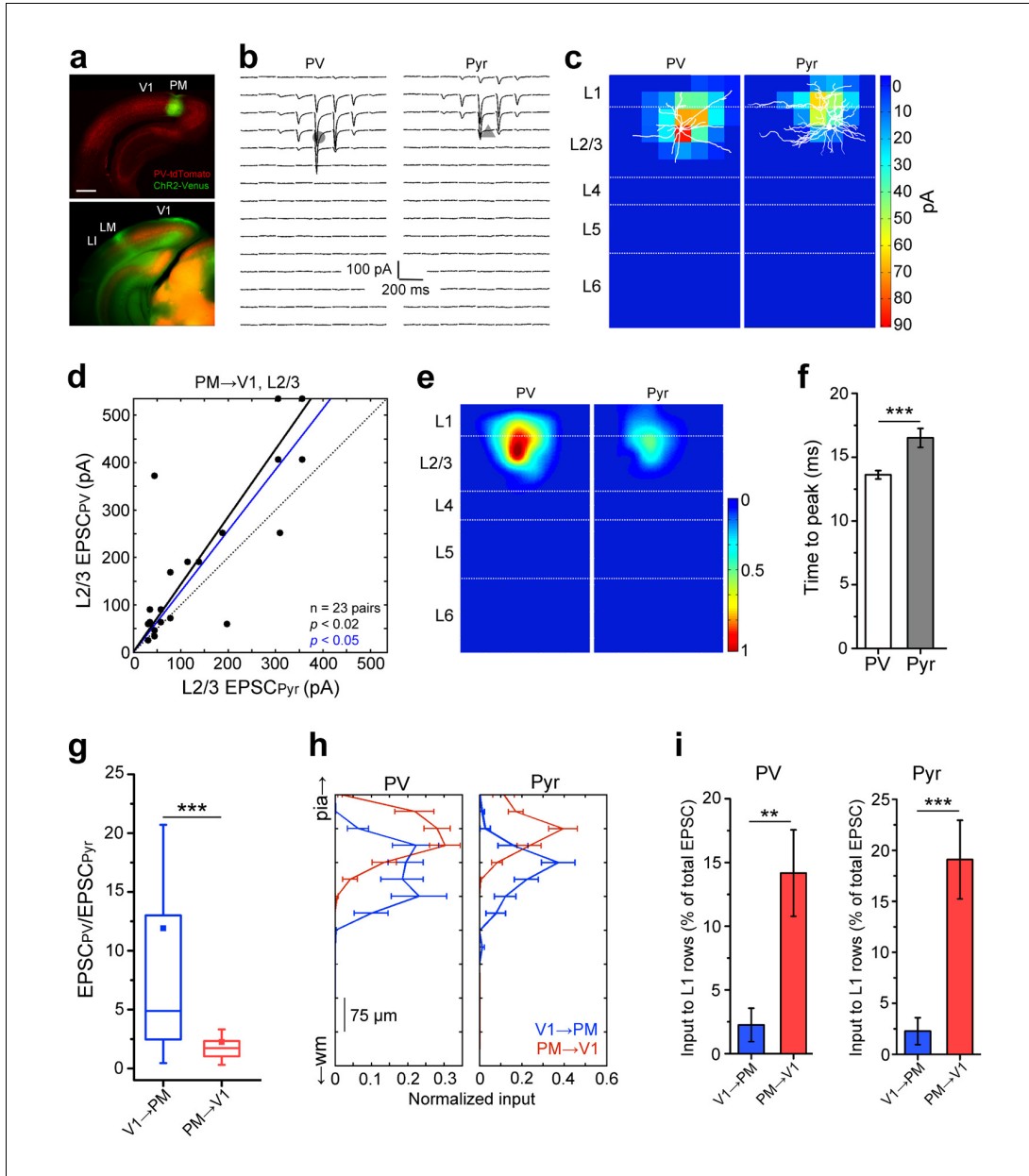

**Figure 3.** Lower I/E balance in PM→V1 pathway. (**a**) Coronal slices showing AAV2/1.CAG.ChR2-Venus (green) injection in PM (top) and axonal labelling in target areas (bottom) of a PV-tdT (red) mouse. Scale bar, 1 mm. (**b**) EPSCs$_{sCRACM}$ in a pair of neighboring PV (left) and Pyr cells (right) in V1. (**c**) Heat map of the currents in *4b* superimposed with biocytin-filled neurons (white). Note significant input into L1 of both cell types. (**d**) PV cells, on average, exhibit larger EPSCs$_{sCRACM}$ than neighboring Pyr cells in the PM→V1 pathway (p<0.02, Wilcoxon signed-rank test). Solid black line: mean slope of data points; blue line: mean slope after normalization to cell conductance. (**e**) Normalized, mean heat map of all L2/3 pairs in the FB$_{PM→V1}$ pathway. (**f**) EPSCs are faster in L2/3 PV than in neighboring Pyr cells upon stimulation of FB$_{PM→V1}$ axon terminals (*p<0.001, paired t-test). (**g**) The interareal excitation of PV cells, normalized to that of neighboring Pyr cells, is on average larger in the FF$_{V1→PM}$ than in the FB$_{PM→V1}$ pathway (***p<0.001, Mann-Whitney U-test). (**h**) Total currents in each row of the 8 × 16 grid for FF$_{V1→PM}$ and FB$_{PM→V1}$ pathways plotted against relative position of each of the 16 rows. EPSCs normalized to total EPSC in each cell-type. pia, pia mater; wm, white matter. (**i**) Interareal input to L1 is stronger in the FB$_{PM→V1}$ than in the FF$_{V1→PM}$ pathway in both cell types (**p<0.01, ***p<0.001, Mann-Whitney U-test). L1 input was calculated as the mean of the total input to each row of the 8 × 16 grid that resided in L1, presented as the percentage of the total input to the neuron.

*Figure 3 continued on next page*

*Figure 3 continued*

The following figure supplement is available for figure 3:

**Figure supplement 1.** Analyses of inputs to L2/3 PV and Pyr neurons in the reciprocal pathways between V1 and PM.

must be noted, however, that due to dendritic filtering of signals, EPSCs at distal dendrites are attenuated more than those near the soma. Thus, the proportion of L1 inputs to the total current may be an underestimate.

## L2/3 FF and FB pathways between LM and PM

Do connections originating from higher areas follow the same normalization rules as those from V1? We addressed this question with sCRACM experiments in L2/3 of $FF_{LM \to PM}$ and $FB_{PM \to LM}$ pathways (*Figure 4*). Similar to FF and FB pathways between V1 and PM, EPSCs were larger in PV cells than in Pyr cells in both pathways (*Figure 4a–f*). In the $FF_{LM \to PM}$ pathway, the mean total current in PV cells was $3.62 \pm 0.75$-fold larger than in neighboring Pyr cells (n = 15 pairs, p<0.02; *Figure 4c,g*). Inputs to both cell types were maximal at proximal dendrites in L3 and 4, but weak in L1 and L2 (*Figure 4a, b,i*; *Figure 4—figure supplement 1*). In the $FB_{PM \to LM}$ pathway, the mean total EPSC to PV cells was $3.77 \pm 0.81$-fold the total EPSC in neighboring Pyr cells, with substantial input into L1 (n = 18 pairs, p<0.01; *Figure 4d–f,h,i*; *Figure 4—figure supplement 1*). Similar to connections between V1 and PM, EPSCs were faster in PV than in Pyr cells in both $FF_{LM \to PM}$ and $FB_{PM \to LM}$ pathways (*Figure 4— figure supplement 2d*). Hence, the faster activation of PV cells appears to be a general rule for FFI provided by long-range connections (*Hull et al., 2009*; *Povysheva et al., 2006*).

While PV cells received stronger excitatory inputs than Pyr cells in all four L2/3 pathways described here, the difference in the relative excitation of PV and Pyr was bigger in the $FF_{V1 \to PM}$ originating at the bottom and terminating at the top of the hierarchy than in the $FF_{LM \to PM}$ pathway originating from the higher area, LM (*Figure 4g*). In contrast, in the $FB_{PM \to V1}$ pathway, the difference was smaller for connections originating at the top and terminating at the bottom of the hierarchy than for terminations at an intermediate level in LM (*Figure 4h*). These relationships are evident in a significant (p<0.001, Kruskal-Wallis test) decrease of the $EPSC_{PV}/EPSC_{Pyr}$ ratios, when pathways are ordered by hierarchical distance from the most feedforward to the most feedback (*Figures 1e*, *4j*). The plot suggests that inhibitory counterbalance to long-range excitation is gradually adjusted depending on the hierarchical location of the source and target areas. Although the total input to PV and Pyr cells differed across pathways, the pathway-specific normalization was independent of the absolute strength of the excitatory input so that the $EPSC_{PV}/EPSC_{Pyr}$ ratios, and not the absolute values of EPSCs in PV and Pyr cells, show a hierarchy-dependent variation (*Figure 4— figure supplement 2a–c*).

## FF and FB pathways in L5

We next asked if interareal inputs to L5 neurons follow a similar physiological connectivity rule as those to L2/3. Unlike in L2/3, the EPSCs recorded in L5 PV and Pyr cells upon stimulation of $FF_{V1 \to PM}$ and $FF_{LM \to PM}$ axons were not significantly different (*Figure 5a–f*). The relative excitation of L5 PV cells, expressed by the $EPSC_{PV}/EPSC_{Pyr}$ ratio, was smaller than that observed in L2/3 for both FF pathways (*Figure 5g,h*). While we did not observe EPSCs in L1 for L5 Pyr cells, likely due to attenuation of signals by dendritic filtering, we detected significant input to L2-4. In particular, L5 Pyr cells in $FF_{LM \to PM}$ exhibited large EPSCs at apical dendrites in L2-4, hundreds of microns distal to the cell body (*Figure 5d,e,i,j*). The proportion of such L2-4 inputs to the total EPSC was higher in $FF_{LM \to PM}$ than in $FF_{V1 \to PM}$ for L5 Pyr cells but not for PV cells, whose input distributions were similar in both pathways (*Figure 5i,j*). Thus depending on the source of long-range synaptic input, L5 Pyr cells in PM receive FF input at different locations of their dendritic arbor.

Finally, we examined the two FB pathways projecting from PM to L5 neurons in V1 and LM respectively. Activation of either $FB_{PM \to V1}$ or the $FB_{PM \to LM}$ projecting axons resulted in EPSCs of similar magnitudes in neighboring PV and Pyr cells (*Figure 6a–f*), with the strongest inputs primarily recorded at proximal dendrites in L5 for both cell types (*Figure 6g*). These results suggest that the

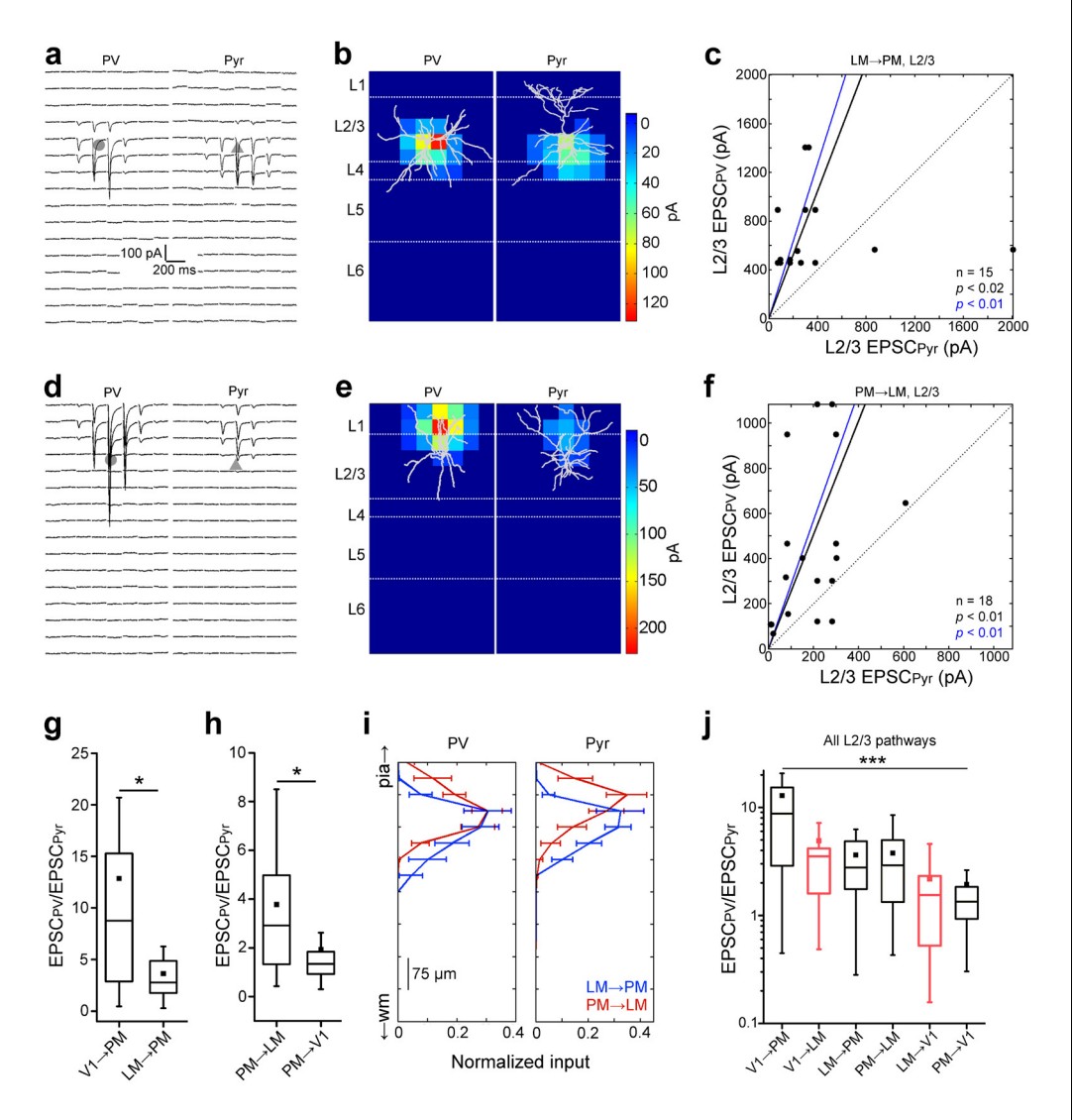

**Figure 4.** Interareal recruitment of L2/3 PV cells depends on the pathway and the hierarchical distance between areas. (a) EPSCs$_{sCRACM}$ in a L2/3 PV (left) and Pyr (right) cell in the FF$_{LM→PM}$ pathway. (b) sCRACM map of EPSCs in *5a* with reconstructed neuron positions. (c) Scatter plot of all PV-Pyr cell pairs in the L2/3 FF$_{LM→PM}$ pathway. PV cells exhibit larger currents than Pyr cells. (d–f) Similar data as in (a–c) but for the L2/3 FB$_{PM→LM}$ pathway. Note stronger L1 input in this pathway. (g) PV cell excitation, normalized to that of a neighboring Pyr cell, is stronger in the FF$_{V1→PM}$ than in the hierarchically shorter FF$_{LM→PM}$ pathway (*p<0.05, Mann-Whiteney U-test). (h) Normalized PV cell excitation is stronger in the FB$_{PM→LM}$ than in the hierarchically longer FB$_{PM→V1}$ pathway (*p<0.05, Mann-Whiteney U-test). (i) Normalized plot of the total current in each row of the 8 × 16 grid, plotted against row position. EPSCs normalized to total current in each cell-type. (j) The total EPSCs$_{sCRACM}$ in a PV cell, normalized to the total EPSCs$_{sCRACM}$ in a neighboring Pyr cell, depends on the directionality of the pathway and hierarchical distance between areas. Red boxes represent data describing connections between V1 and LM from *Yang et al. (2013)*. ***p<0.001, Kruskal-Wallis test.

The following figure supplements are available for figure 4:

**Figure supplement 1.** Stronger input to L1 in the PM-to-LM than in the LM-to-PM pathway.

**Figure supplement 2.** Comparisons of strengths, extent, and rise times of inputs in L2/3 pathways.

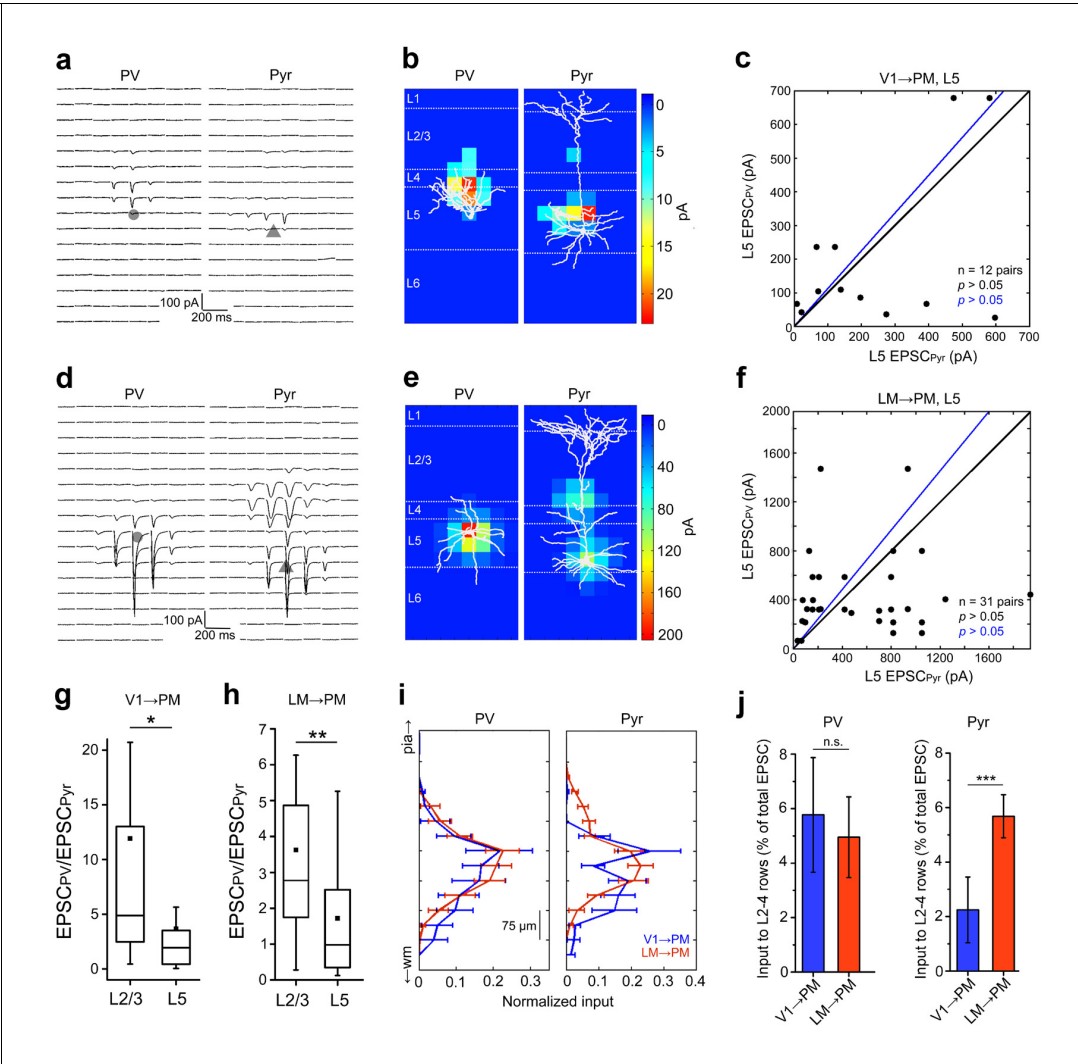

**Figure 5.** FF input to L5 neurons. (**a**) FF$_{V1\rightarrow PM}$ EPSCs$_{sCRACM}$ in a pair of neighboring L5 PV (left) and Pyr (right) cells. (**b**) Heat map of EPSCs in *6a* superimposed with respective biocytin-filled L5 neurons. (**c**) Scatter plot, as previously described, of EPSCs$_{sCRACM}$ in PV and Pyr cell pairs in L5 FF$_{V1\rightarrow PM}$. The total current in PV and Pyr cells were not significantly different. (**d–f**) Similar data as in *Figure 6a—c* but for L5 FF$_{LM\rightarrow PM}$. (**g,h**) PV cell excitation, normalized to the excitation of a neighboring Pyr cell, is stronger in L2/3 than in L5 for both V1→PM (**g**) and LM→PM pathways (**h**). (**i**) Total EPSC in each row of the 8 × 16 grid normalized to total current recorded, plotted against row position (16 rows). Note that L5 PV cells do not show significant differences in the laminar distribution of EPSCs in the two pathways, but L5 Pyr cells receive more input in the upper layers from LM than from V1. (**j**) Interareal input in L2-4 for L5 PV cells (left) in PM is not significantly different for the two pathways. L5 Pyr cells (right) receive more L2-4 input in the FF$_{V1\rightarrow PM}$ than in the FF$_{LM\rightarrow PM}$ pathway. L2-4 input calculated as the average EPSC in each row that resided in L2-4, shown as the percentage of the total EPSC in the cell (***p<0.001, Mann-Whitney U-test).

stronger activation of PV cells observed in L2/3 is absent in L5. Consistent with this observation, we found no significant difference between the EPSC$_{PV}$/EPSC$_{Pyr}$ ratios for L5 cell pairs for the different pathways (*Figure 6h*), suggesting equal potency of FFI among these pathways regardless of whether they are FF or FB. Similar to L2/3, however, the EPSCs in PV cells showed faster rise times than those in Pyr cells in all L5 pathways (*Figure 6i*).

## Discussion

We have mapped input strengths to inhibitory PV and excitatory Pyr cells in diverse pathways interconnecting three visual cortical areas with distinct spatiotemporal sensitivities and specialized

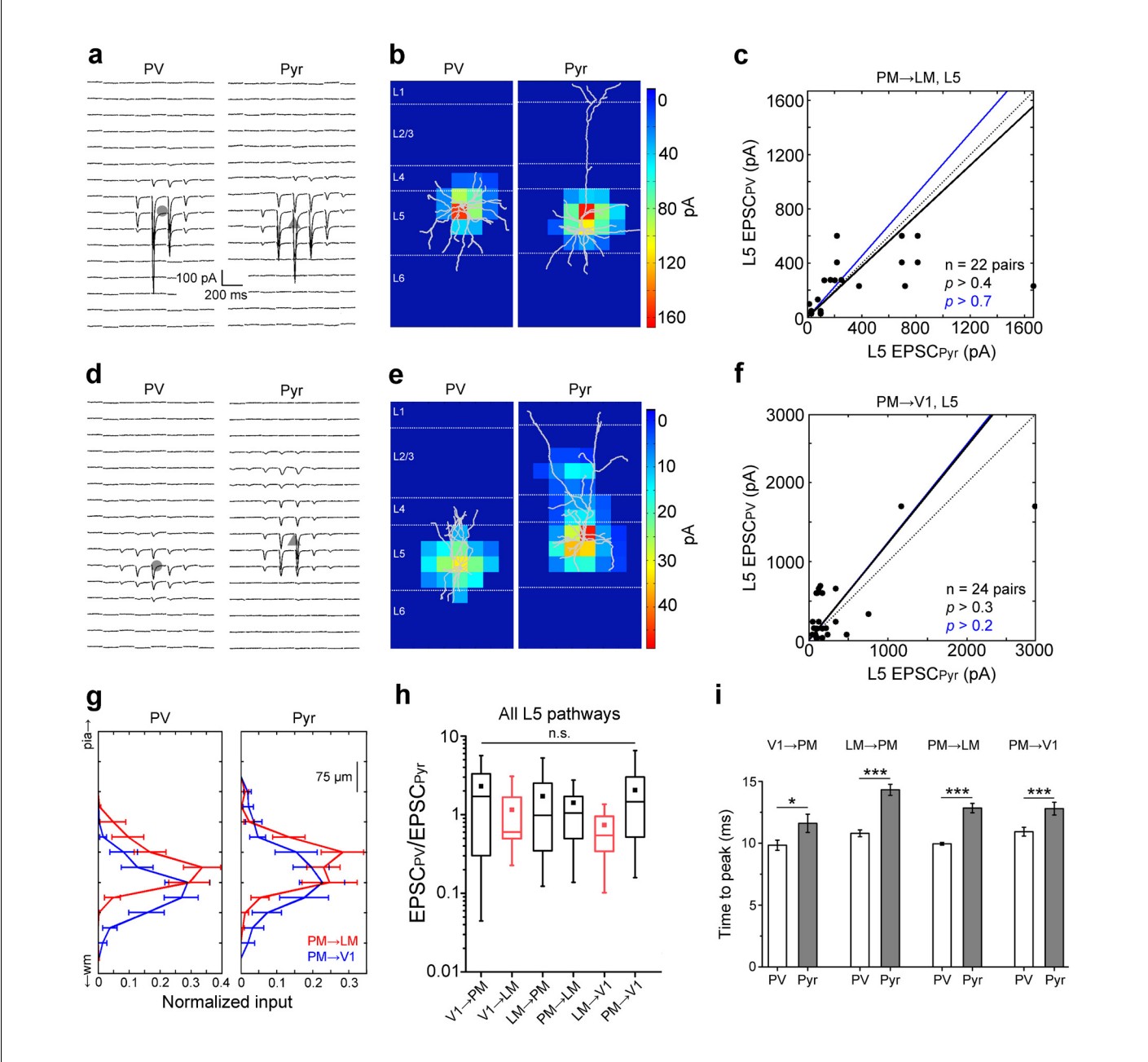

**Figure 6.** FB input to L5 neurons. (a) FB$_{PM \to LM}$ EPSCs$_{sCRACM}$ in a pair of neighboring L5 PV (left) and Pyr (right) cells. (b) Heat map of EPSCs from *7a* superimposed with the respective biocytin-filled L5 neurons. (c) Scatter plot of all L5 PV-Pyr neuron pairs receiving input from FB$_{PM \to LM}$. Total EPSC in the two cell types are not significantly different. (d–f) Similar data as *7a-c* but for the FB$_{PM \to V1}$ pathway. (g) Total EPSC in each row of the stimulation grid plotted against row position. The grids of the two different pathways are aligned to pial surface. (h) EPSCs in PV cells normalized to EPSCs in neighboring Pyr cells (EPSC$_{PV}$/EPSC$_{Pyr}$) for all L5 pathways arranged from most FF to most FB. Unlike in L2/3, the EPSC$_{PV}$/EPSC$_{Pyr}$ ratios in L5 are not significantly different in different pathways (p>0.2, Kruskal-Wallis test). Red boxes describe data from *Yang et al. (2013)*. (i) Interareal EPSCs are faster in PV than in Pyr cells in all L5 pathways (*p<0.05, ***p<0.001, paired t-test).

functions (*Marshel et al., 2011*; *Andermann et al., 2011*; *Roth et al., 2012*; *Glickfeld et al., 2014*). The results in L2/3 support the notion that in FF and FB pathways, excitation is more strongly counterbalanced by inhibition and that the imbalance is gradually rectified according to hierarchical distance from the most FF to the most FB (*Figure 1e*, *4j*). The results further suggest that the

hierarchical distance rule of normalization is independent of the absolute magnitude of EPSCs across the hierarchy (*Figure 4—figure supplement 2a–c*). Our findings argue that excitation ascending across multiple hierarchical levels is gradually adjusted to keep the dynamic range of L2/3 Pyr cell firing constant and compensate for the increased density of synaptic input to Pyr cells in higher cortical areas (*Elston, 2003*). Strong activation of PV neurons may narrow the window for effective excitation and result in high frequency gamma-band synchronization of activity found in FF signaling (*Gabernet et al., 2005*; *Cardin et al., 2009*; *Bastos et al., 2015*). In contrast, in FB pathways excitation is weakly counterbalanced by inhibition, which may broaden the window for synaptic integration and result in slower synchronization frequencies found in FB communications (*Bastos et al., 2015*). Thus, variation in I/E balance, through the differential recruitment of PV and Pyr neurons in different cortical pathways, is a key feature of distributed hierarchical processing.

Reciprocal connections between areas are a highly conserved feature of the mammalian cortex. However, the exact pattern of termination of FF and FB axonal projections in the target area appears to vary between species, particularly in the termination patterns of FF pathways in layers 2, 3 and 4 (*Felleman and Van Essen, 1991*; *Price and Zumbroich, 1989*; *Coogan and Burkhalter, 1990*). Despite these differences, a consistent observation among different species is a tendency for FF projections to avoid L1 and the selective targeting of L1 by FB pathways (*Coogan and Burkhalter, 1993*; *Rockland and Virga, 1989*; *Henry et al., 1991*; *Cauller, 1995*). We therefore used the average DR of axonal terminations in L2-4 to those in L1 to classify pathways on a sliding scale as being FF or FB. In this reference frame, V1, LM and PM constitute a clear hierarchy, which broadly matches that of rat visual cortex (*Coogan and Burkhalter, 1993*) and is consistent with the increasing size of receptive fields (*Wang and Burkhalter, 2007*). The hierarchical ordering of V1, LM and PM based on average DRs is consistent with the ordering based on the difference of DRs between reciprocally connected pairs. This is notable because differences in the laminar patterns of reciprocal projections between two areas have traditionally been used to arrange areas in a hierarchy (*Felleman and Van Essen, 1991*; *Coogan and Burkhalter, 1993*). While our method of averaging DRs provides a hierarchy based on how individual visual areas project to every other area within the network, it is conceivable that such a hierarchical arrangement may not be consistent with defining pathways between every reciprocally connected pair of areas as being FF or FB by comparing the DRs of projections to each other. The absolute value of the difference between DRs of reciprocally connected areal pairs therefore remains an open issue for defining hierarchical distance and designating connections as FF, FB, or lateral (*Felleman and Van Essen, 1991*; *Coogan and Burkhalter, 1993*).

Cortical Pyr cells typically receive thousands of synaptic contacts, raising the question of how these neurons successfully generate graded spike outputs, without saturating their spike output, in response to varying levels of excitatory input (*Shadlen and Newsome, 1998*). This problem is compounded by the need for deeper parts of the brain, which are further separated from the outside world than lower areas, to respond robustly and appropriately to sensory input varying in intensity over several orders of magnitude. Pertinently, Pyr cells in higher areas have been shown to have a higher density of dendritic spines than those in lower areas in both primates (*Elston, 2003*) and rodents (*Elston et al., 2006*), indicating that Pyr neurons in higher areas must integrate a larger number of excitatory inputs. To maintain a wide dynamic range over which Pyr cells can signal, inhibitory neurons have been proposed to be critical (*Shadlen and Newsome, 1998*; *Pouille et al., 2009*). In particular, PV neurons normalize cortical activity by inhibiting Pyr cells by a level that is proportional to the latter's excitation, thus controlling their gain (*Atallah et al., 2012*; *Wilson et al., 2012*; *Xue et al., 2014*). Because they are strongly targeted by interareal inputs (*Gonchar and Burkhalter, 1999*), PV cells are also ideally suited to mediate long-range FFI between areas. Such an interareal inhibitory circuit would make Pyr cells coincidence-detectors (*Gabernet et al., 2005*; *Pouille and Scanziani, 2001*), leading to a reduction of noise levels and the preservation of temporal precision in the target area (*Bruno, 2011*; *Zhu et al., 2015*). Coincidence-detection has also been proposed to help achieve a wide dynamic range by allowing only a fraction of excitatory inputs to summate and evoke a spike response (*Shadlen and Newsome, 1998*). Our observation that L2/3 PV cells are recruited most strongly by pathways transmitting signals from V1 to higher cortical areas imply that signals sent to deeper parts of the brain from more peripheral areas are more potently controlled by inhibition than pathways originating in higher areas. Such a high level of inhibition may be crucial in order for Pyr cells to efficiently integrate excitatory input from a large number of areas.

On the other hand, lower I/E levels in FB pathways would broaden the 'window of opportunity' for spikes to be integrated and trigger an output in the postsynaptic cell (*Isaacson and Scanziani, 2011*), suggesting that FB signals originating in association cortex require less gain control than FF signals. Rather, by broadly modulating the excitability of neurons in lower areas (such as by targeting the primary dendrites of Pyr cells in L1/2), FB pathways are well-placed to prime Pyr cells to selectively respond to FF input in a context-dependent manner (*Larkum, 2013*).

Although synaptic inputs to L5 Pyr cells are also denser in higher areas (*Elston and Rosa, 2000*), we found that excitation of these neurons in FF and FB pathways is similar and appears to be less strongly counterbalanced by inhibition. This provides a putative mechanism for the previously observed sparse coding in L2/3 Pyr cells and dense excitation in intrinsically burst-spiking L5 Pyr cells, allowing for distinct computational strategies within individual neurons depending on their postsynaptic targets (*Harris and Mrsic-Flogel, 2013*; *Hefti and Smith, 2000*; *Schubert et al., 2007*). The laminar difference may indicate that, similar to thalamocortical input (*Constantinople and Bruno, 2013*), interareal inputs to L5 are driving Pyr cells. This may enable interareal communication through cortico-thalamo-cortical loops (*Guillery and Sherman, 2002*) as well as with subcortical motor targets, thereby linking perception and action (*Kim et al., 2015*).

While PV neurons are a critical component of cortical gain control, it must be noted that they are only one of a number of inhibitory sources (*Jiang et al., 2015*; *Pfeffer et al., 2013*; *Gonchar et al., 2007*). For instance, neocortical GABAergic interneurons that express vasoactive intestinal polypeptide (VIP) are thought to be an important target of long-range and neuromodulatory inputs (*Fu et al., 2014*; *Reimer et al., 2014*), and in turn, primarily inhibit other interneurons leading to disinhibition of cortical Pyr cells (*Pfeffer et al., 2013*; *Kepecs and Fishell, 2014*; *Pi et al., 2013*; *Karnani et al., 2016*). Somatostatin (SOM)-expressing interneurons, which include Martinotti cells, make extensive inhibitory contacts with local Pyr cells, and can consequently mediate disynaptic inhibition between neighboring Pyr cells (*Fino and Yuste, 2011*; *Silberberg and Markram, 2007*). SOM neurons have also been shown to provide inhibitory inputs to other interneurons, including PV cells, suggesting a role in the disinhibition of Pyr cells as well (*Pfeffer et al., 2013*). A perhaps surprising source of inhibition and disinhibition is glutamate. By activating pre- and postsynaptic metabotropic receptors in various neocortical circuits, glutamate release can induce suppression of GABA release and inhibition of L4 neurons, respectively (*Liu et al., 2014*; *Lee and Sherman, 2009*). Thus, multiple, partially overlapping (*Gonchar et al., 2007*) sources of inhibition may be differentially recruited depending on context, providing a multilayered control of cortical function (*Kepecs and Fishell, 2014*; *Pakan et al., 2016*).

## Materials and methods

All experimental procedures were approved by the Institutional Animal Care and Use Committee at Washington University.

### Animals

For analyzing projection patterns between cortical areas, we used 6–8 week-old C57BL/6J male and female mice. In addition, we crossed *Pvalb-Cre* mice (RRID:IMSR_JAX:008069) with Ai9 reporter mice (C57BL/6 background, The Jackson laboratory, Bar Harbor, ME; RRID:IMSR_JAX:007905), which harbored a floxed STOP cassette that prevents transcription of the fluorescent protein tdTomato (tdT). The crossing resulted in offspring in which PV neurons express tdT. All electrophysiology experiments were performed in male and female PV-tdT mice.

### Tracing connections

Mice were anesthetized by intraperitoneal injection of a ketamine/xylazine (86 mg·kg$^{-1}$/13 mg·kg$^{-1}$, IP) mixture and secured in a headholder. Analgesia was achieved by buprenorphine (5 mg·kg$^{-1}$, SC). Callosal connections were labeled by 30–40 pressure injections (20 nl each) of the retrograde tracer bisbenzimide (BB, 5% in H$_2$O, Sigma) into the right occipital cortex. Interareal projections were labeled by iontophoretic injections (3 µA, 7 s on/off duty cycle for 7 min) of the anterograde tracer biotinylated dextran amine (BDA; 10,000 molcular weight, 5% in H$_2$O; Invitrogen) using a coordinate system whose origin was the intersection between the midline and a perpendicular line drawn from the anterior border of the transverse sinus at the posterior pole of the occipital cortex. The

coordinates of the injected areas were (anterior/lateral in mm): V1 (1.1/2.6); LM (1.4/4.1); PM (1.9/1.6). Mice were randomly assigned for injections of a particular area.

## Visualization of connections

Three days after the tracer injections, the mice were overdosed with ketamine/xylazine, perfused through the heart with heparinized phosphate buffer (PB; 0.1 M, pH 7.4) followed by 4% paraformaldehyde in PB (PFA). Brains were postfixed with 4% PFA and equilibrated in 30% sucrose. To enable areal identification of injection and projection sites, BB labeled callosal landmarks in the left hemisphere were imaged in situ under a fluorescence stereomicroscope (Leica MZ16F), equipped with UV optics. The imaged hemispheres were then cut on a freezing microtome at 40 μm in the coronal plane. Sections were collected and numbered as a complete series across the full caudo-rostral extent of the hemisphere. Sections were wet mounted onto glass slides and imaged under UV illumination under a fluorescence microscope equipped with a CCD camera. The sections were then removed from the slides and BDA labeled axonal projections were visualized with avidin and biotinylated HRP (Vectastain ABC Elite) in the presence of $H_2O_2$ and diaminobenzidine (DAB) (*Wang et al., 2012*). Sections were mounted onto glass slides, coverslipped in DPX and imaged under a microscope equipped with dark field optics.

## Virus injections

16 to 23-day-old mice were anesthetized with a mixture of ketamine/xylazine (86 mg·kg$^{-1}$/13 mg·kg$^{-1}$, IP). Held in a stereotaxic apparatus, intracerebral injections of viral vector (AAV2/1.CAG. ChR2-Venus.WPRE.SV40 (Addgene20071); Vector Core, University of Pennsylvania) (*Petreanu et al., 2009*) were made with glass pipettes (tip diameter 25 μm) connected to a Nanoject II Injector (Drummond). Injections were performed stereotaxically into V1, LM or PM, 0.3 and 0.5 mm below the pial surface, to ensure infection of neurons throughout the thickness of cortex. The total volume of the viral vector at each depth was 46 nl. Successful injections resulted in the simultaneous expression of Channelrhodopsin-2 (ChR2) and the fluorescent protein Venus in terminals of outgoing axons. Mice were randomly selected for the study of a particular pathway.

## Slice electrophysiology

30 to 45 day-old mice, 14–21 days after viral injection, were anesthetized with a mixture of ketamine/xylazine (86 mg·kg$^{-1}$/13 mg·kg$^{-1}$, IP), and transcardially perfused with 10 ml of ice-cold oxygenated 95% $O_2$/5% $CO_2$ dissection solution (sucrose-ACSF) containing (in mM): 228 sucrose, 2.5 KCl, 1.25 $NaH_2PO_4$, 25 $NaHCO_3$, 0.5 $CaCl_2$, 7.0 $MgCl_2$, and 10 D-glucose. Mice were decapitated, the brain removed from the skull, and mounted on the specimen plate of Leica Vibratome (Leica VT1200) with a cyanoacrylate adhesive (Krazy Glue). Visual cortex was cut coronally at 350 μm in ice-cold sucrose-ACSF. Slices were transferred to a holding chamber filled with ACSF containing (in mM): 125 $NaCl_2$, 2.5 KCl, 1.25 $NaH_2PO_4$, 25 $NaHCO_3$, 2.0 $CaCl_2$, 1.0 $MgCl_2$, and 25 D-glucose. Slices were incubated in ACSF for 30 min at 34°C and maintained at room temperature until recordings. Acute slices were superfused with recirculating oxygenated ACSF at room temperature in a submersion chamber mounted on the fixed stage of an upright microscope (Nikon Eclipse FN1). For subcellular, optogenetic mapping experiments, 1 μM TTX and 100 μM 4-AP were added to the bath in order to block action potentials (and therefore polysynaptic excitation) and fast repolarizing potassium currents. Whole-cell patch clamp recordings were performed with borosilicate pipettes (4–6 MΩ resistance). The pipette solution contained (in mM) 128 potassium gluconate, 4 $MgCl_2$, 10 HEPES, 1 EGTA, 4 $Na_2ATP$, 0.4 $Na_2GTP$, 10 sodium phosphocreatine, 3 sodium L-ascorbate, and 3 mg/ml biocytin. The pH was adjusted to 7.2–7.3, and osmolarity to 290 mOsm. Fluorescence of ChR2/Venus-expressing fibers and tdT-expressing PV neurons was imaged with a CCD camera (Retiga 2000DC, QImaging). Pyr and PV neurons lying within maximal levels of ChR2/Venus-expressing axonal projections were selected for recordings. PV neurons were identified by tdT expression. For sCRACM experiments (see below), neurons were voltage clamped at −70 mV. All voltage-clamp and current-clamp experiments were performed using the Ephus software (*Suter et al., 2010*) (Vidrio Technologies), an Axopatch 700B amplifier (Molecular devices), and a data acquisition (DAQ) device (NI USB-6259, National Instruments Corp., Austin, TX).

## Optogenetic photostimulation

The photostimulation of ChR2-expressing fibers was achieved by a blue laser (473 nm; CrystaLaser) delivered in an 8 × 16 grid in which stimulation points were spaced 75 µm apart, one spot at a time, 400 ms between laser delivery at each spot. The grid was aligned such that the longer axis was perpendicular to the pial surface and stimulated spots in all six layers. The position of the laser beam was controlled by galvanometer scanners (Cambridge Scanning), and the duration of stimulation (1 ms) was controlled by a shutter (LS6, Uniblitz). The laser beam (~20 µm at half maximal intensity) passed through a Pockels cell (ConOptics) and an air objective (4x PlanApo). Because the expression level of ChR2-Venus in interareal axons varied across slices and animals, the laser power was adjusted in every slice so that the largest $EPSC_{sCRACM}$ (EPSC recorded under sCRACM conditions) in a neuron did not increase upon increasing laser intensity. Importantly, the laser power was constant for all recordings made in the same slice, in order to compare $EPSCs_{sCRACM}$ between neighboring neurons. The laser power measured at the image plane was 0.7–1 mW/cm (2). Photostimulation was repeated three to five times for each neuron. The shutter timing and the position of galvanometer mirrors was controlled by Ephus (*Suter et al., 2010*).

## Immunostaining

After the recordings, slices were fixed in 4% PFA, cryoprotected in 30% sucrose and re-sectioned on a freezing microtome at 40 µm. The sections were then incubated with an antibody against the type 2 muscarinic acetylcholine receptor (M2; 1:500 in PB; MAB367, Millipore; RRID:AB_94952) and stained with Alexa Fluor 647-labeled IgG (1:500 in 10% NGS; A21247; Invitrogen). M2-expression was imaged under a microscope equipped with IR fluorescence optics. The intense M2-expression in V1 was used as a landmark for assigning Venus labeled axonal projections to LM and PM (*Wang et al., 2011*).

## Dendritic morphology

M2 stained sections containing biocytin-filled neurons were treated with 1% $H_2O_2$, and incubated in avidin and biotinylated horseradish peroxidase (Vectastain ABC Elite) in the presence of DAB. The soma and dendritic arbor of biocytin-filled neurons were reconstructed under a 60x oil objective using Neurolucida (MBF Bioscience; RRID:SCR_001775).

## Data analysis and statistics

### Areal hierarchy analysis

One BDA injection was performed in each mouse, and injection into a particular area (V1, LM, or PM) was performed in two mice (n = 6 animals for all injections). Three adjacent sections containing projections in the target area were typically used for analyses of each pathway in each brain. Projections were assigned to areas by their location relative to retrogradely bisbenzimide-labeled callosal landmarks (*Wang and Burkhalter, 2007*) and by their relative positions to each other. Callosal landmarks were imaged in situ before sectioning the brain. Sections were numbered from the posterior pole of cortex so that the callosal landmarks seen in the coronal plane could be matched to specific locations (multiplying the section number by the section thickness) of the in situ pattern. BDA labeled injection sites and axonal projections were then superimposed onto the callosal pattern observed in the same section, and terminations were assigned to specific areas according to the map by Wang and Burkhalter (*Wang and Burkhalter, 2007*).

Grayscale images of anterogradely BDA-labeled axonal projections in target areas were used for analyses of termination patterns. The coronal sections were imaged under 8x magnification. A custom-written MATLAB script were used to generate contour plots of the optical density of axons after processing the image through a circular averaging 2-D filter. Previous analyses have shown that optical density correlates with bouton density (*Wang et al., 2011*). Regions within the contours of the highest 70% of optical densities in L2-4 and in L1 were used to generate the L2-4:L1 ratio for each slice. The optical density was measured using the mean *Gray Value* in ImageJ (RRID:SCR_003070) within the 70% contour.

## Electrophysiology analyses

EPSCs recorded upon photostimulation, >4 times the standard deviation of baseline, were used for analysis. Individual pixel values for each position of the $8 \times 16$ photostimulus grid was calculated as the average EPSC value within 75 ms after photostimulation, and expressed in pA. These calculations were done by custom-written MATLAB scripts. EPSCs at each location of the grid were averaged over three to five repetitions of photostimulation to generate sCRACM maps for each neuron. To compare the total interareal input to pairs of PV and Pyr neurons in the target area, we summed the pixel values for each cell, and compared the total EPSC value of PV and Pyr neurons lying within ~100 µm of each other, either in L2/3 or in L5. For comparison of mean EPSCs per stimulation point (*Figure 3—figure supplement 1*), we averaged pixel values with significant responses (>4 times standard deviation of baseline) for each cell. For statistical analysis of differences of interareal input to PV and Pyr cells for a particular pathway, we generated scatter plots (for example, *Figure 2f*) in which each data point plotted the total $EPSC_{sCRACM}$ from a PV neuron (vertical axis) against the total $EPSC_{sCRACM}$ from its Pyr neighbor (horizontal axis). The solid black line in such a scatter plot was generated by connecting the origin (0, 0) to the geometric mean of all $EPSC_{PV}/EPSC_{Pyr}$ ratios for the respective pathway. The solid blue line was plotted in a similar fashion, but after normalization to the average cell conductance. The non-parametric Wilcoxon signed-rank test was used for comparing total EPSCs between cell types within pairs. For average heat maps of multiple PV or Pyr neurons, we used a smoothening function in MATLAB that interpolates EPSC values between pixels.

All box plots show mean (black squares), median (horizontal line within box), 25–75 percentile range (horizontal lines bounding box) and outermost points within upper and lower inner fences (whiskers). The non-parametric Kruskal-Wallis test was used to compare mean $EPSC_{PV}/EPSC_{Pyr}$ ratios between pathways, while the One-Way Analysis of Variance (ANOVA) was used for comparing the means of groups whose probability distributions were expected to be parametric. Statistical significance was $p < 0.05$. No statistical method was used to predetermine the number of neurons, slices, or animals used (sample size), but our sample sizes were consistent with other comparable experiments (*Wang et al., 2012*; *Yang et al., 2013*; *Mao et al., 2011*).

## Acknowledgements

We thank Katia Valkova and Weiguo Yang for technical assistance. Work supported by R01 EY016184, R01 EY022090, and the McDonnell Center for Systems Neuroscience.

## Additional information

### Funding

| Funder | Grant reference number | Author |
| --- | --- | --- |
| National Eye Institute | R01 EY016184 | Andreas Burkhalter |
| McDonnell Center for Systems Neuroscience | | Andreas Burkhalter |
| National Eye Institute | R01 EY022090 | Andreas Burkhalter |

The funders had no role in study design, data collection and interpretation, or the decision to submit the work for publication.

### Author contributions

RDD, Performed optogenetic and electrophysiological experiments, Conception and design, Acquisition of data, Analysis and interpretation of data, Drafting or revising the article; AMM, Designed and performed analyses of electrophysiology data, Analysis and interpretation of data; PB, Assisted in setting up and in performing pilot paired recordings experiments, Contributed unpublished essential data or reagents; QW, Performed BDA tracing experiments, Acquisition of data; AB, Performed BDA tracing experiments, Conception and design, Acquisition of data, Drafting or revising the article

## Author ORCIDs

Rinaldo David D'Souza, http://orcid.org/0000-0001-9028-1990
Andreas Burkhalter, http://orcid.org/0000-0002-5140-3657

## Ethics

Animal experimentation: All experimental procedures were approved by the Institutional Animal Care and Use Committee at Washington University (protocol numbers 20130104 and 20160093) and conformed to guidelines set by the National Institutes of Health.

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
