## [Decision Letter]

Thank you for submitting your article "Pathway-specific recruitment of inhibition and excitation across the counterstream mouse visual cortical network" for consideration by *eLife*. Your article has been favorably evaluated by Andrew King as the Senior Editor and three reviewers: Sandra Kuhlman (Reviewer #2); Ed Callaway (Reviewer #3), and Sacha B Nelson who is a member of our Board of Reviewing Editors.

The reviewers have discussed the reviews with one another and the Reviewing Editor has drafted this decision to help you prepare a revised submission.

Summary:

This is an interesting and important paper that explores anatomical and functional differences that help to both define and characterize feedforward (FF) versus feedback (FB) corticortical systems in the mouse visual cortex. How the relative recruitment of inhibitory and excitatory neurons is distributed across the visual hierarchy, both in terms of feedforward and top-down feedback is of great importance for constraining models of visual function.

Essential revisions:

1) One of the reviewers was concerned by the absence of normalized input plots for FB connections to L5 neurons. Did this imply technical concerns with these data? If so, these should be discussed. If there are no such concerns it would probably be better to include the plots in Figure 7 to allow comparison with other figures.

2) The reviewers felt that it was important to modify the Discussion to acknowledge the caveat of other sources of inhibition not studied by focusing on parvalbumin positive interneurons.

3) The reviewers felt that the observation of delayed spiking following activation of interneurons added little. If indeed the authors would like to conclude that interareal connections for FFI form local subnetworks, more direct evidence such as anatomical connectivity, is required.

4) Revision of the title – the title is jargony and should be revised.

---

## [Author Response]

*Essential revisions:*

*1) One of the reviewers was concerned by the absence of normalized input plots for FB connections to L5 neurons. Did this imply technical concerns with these data? If so, these should be discussed. If there are no such concerns it would probably be better to include the plots in Figure 7 to allow comparison with other figures.*

We have now included the normalized input plots of FB L5 neurons in the new Figure 6.

*2) The reviewers felt that it was important to modify the Discussion to acknowledge the caveat of other sources of inhibition not studied by focusing on parvalbumin positive interneurons.*

We agree with the reviewers’ comments and have revised the Discussion (last paragraph) by adding text describing the possible contribution of other types of interneurons in the scaling of excitation.

*3) The reviewers felt that the observation of delayed spiking following activation of interneurons added little. If indeed the authors would like to conclude that interareal connections for FFI form local subnetworks, more direct evidence such as anatomical connectivity, is required.*

We consider the data in question (old supplementary Figure 4G-IFigure 4) not an essential part of the story and have therefore decided to eliminate them from the revised manuscript.

*4) Revision of the title – the title is jargony and should be revised.*

We agree that the old title was geared toward a highly specialized audience. We have therefore replaced the old title which now reads: “Recruitment of inhibition and excitation across mouse visual cortex depends on the hierarchy of interconnected areas”.